## RESEARCH ARTICLE

# PGAM5 cleavage and oligomerization equilibrates mitochondrial dynamics under stress by regulating DRP1 function

Sudeshna Nag[1,*], Kaitlin Szederkenyi[1,2], Christopher M. Yip[1,2] and G. Angus McQuibban[1,*]

## ABSTRACT

Mitochondrial dynamics relies on the function of dynamin family GTPase proteins including mitofusin 1 (MFN1), mitofusin 2 (MFN2) and dynamin-related protein 1 (DRP1; also known as DNM1L). The mitochondrial phosphatase phosphoglycerate mutase 5 (PGAM5) protein can regulate the phosphorylation levels and the function of both MFN2 and DRP1; however, the precise regulation of PGAM5 activity is unknown. Here, we show that PGAM5 oligomerization and localization controls its function. Under depolarization and/or metabolic stress PGAM5 changes its association and, instead of forming dodecamers, forms dimers. These PGAM5 oligomers have differential affinity towards MFN2 and DRP1. Simultaneously, PGAM5 is cleaved by the inner mitochondrial membrane-resident proteases PARL and OMA1 and a fraction of the cleaved PGAM5 translocates to the cytosol. These two events play an important role in regulating mitochondrial dynamics under depolarization and/or metabolic stress. Taken together, our results identify PGAM5 oligomerization and cleavage-induced relocalization as crucial regulators of its function.

KEY WORDS: Mitochondrial morphology, PGAM5, MFN2, DRP1, Glucose starvation

## INTRODUCTION

Mitochondria are metabolic hubs crucial for generating ATP, maintaining $Ca^{2+}$ balance and regulating cellular homeostasis, and mitochondrial morphology plays a pivotal role in governing these diverse functions. Recent studies have indicated that alterations in mitochondrial shape and membrane dynamics are associated with aging and development of neurodegenerative conditions, such as Parkinson's disease (PD) and Alzheimer's disease (AD) (Quintana-Cabrera and Scorrano, 2023).

Mitochondrial morphology relies on a balance between membrane fusion and fission. Fusion is governed by mitofusins (MFN1 and MFN2) found in the outer mitochondrial membrane (OMM) and optic atrophy 1 (OPA1) located in the inner mitochondrial membrane (IMM). Fission is regulated primarily by the OMM-localized protein dynamin-related protein 1 (DRP1 also known as DNM1L) (Sprenger and Langer, 2019; Chan et al., 2019; Tilokani et al., 2018). The activity, localization and turnover of these key mitochondrial dynamics proteins is further regulated by post-translational modifications such as phosphorylation and ubiquitylation. Phosphorylation and ubiquitylation of mitofusins enhances fission. By contrast, DRP1 has two phosphorylation sites, which function in opposing ways. Phosphorylation of the serine 637 residue enhances fusion whereas phosphorylation of the serine 616 residue enhances fission. These modifications are synchronized with the upstream conditions, such as cellular stress and metabolic state, and physiological processes, like cell cycle progression, thus defining their functional context (Sabouny and Shutt, 2020).

Recently, phosphoglycerate mutase 5 (PGAM5), a mitochondrial serine/threonine phosphatase, has emerged as a crucial regulator of mitochondrial morphology. Importantly, PGAM5 function is clinically relevant for multiple human diseases (Cheng et al., 2021; Wang et al., 2012). Upregulated PGAM5 expression level is associated with several forms of cancer, including hepatocellular carcinoma (HCC), human colorectal cancer (CRC) and adenocarcinoma (Wang et al., 2012). Elevated PGAM5 protein levels in blood plasma also serves as a bio-marker of PD in individuals over 66 years of age (Feng et al., 2022; Cheng et al., 2021).

PGAM5 regulates mitochondrial dynamics in multiple ways (O'Mealey et al., 2017; Zhang et al., 2018; Nag et al., 2023a; Lu et al., 2014). In mammalian cells, PGAM5 dephosphorylates DRP1 resulting in increased mitochondrial fission. PGAM5–DRP1 interaction initiates programmed necrosis in cervical cancer, colon cancer and prolactinoma cells (Wang et al., 2012; O'Mealey et al., 2017; Nag et al., 2023a). PGAM5 also regulates mitochondrial retrograde trafficking induced by proteosome inhibition (Zhang et al., 2018).

In a previous study, we found that PGAM5 interacts with MFN2 and DRP1 in a stress-responsive manner (Lu et al., 2014). Under steady-state conditions, PGAM5 interacts and regulates MFN2 phosphorylation and enhances fusion. Conversely, under depolarization stress, PGAM5 interacts with DRP1. These contrasting interaction patterns of PGAM5 raise important questions – how is its function regulated, and how does PGAM5 choose its substrates in order to maintain proper mitochondrial dynamics and healthy cellular homeostasis.

In this study, we demonstrate that different PGAM5 association states or oligomerization forms can regulate stress-induced mitochondrial dynamics. The dodecameric form of PGAM5 interacts with MFN2 whereas its dimeric form interacts with DRP1. Our results show that cellular stress stimulates changes in PGAM5 self-association state and its cleavage at the IMM, which enhances its interaction with DRP1.

[1]Department of Biochemistry, University of Toronto, MaRS Centre West Tower, 661 University Ave., M5G 1M1, Toronto, Canada. [2]Terrence Donnelly Centre for Cellular and Biomolecular Research, 160 College Street, M5S 3E1, Toronto, Canada.

*Authors for correspondence (sudeshna.iisc@gmail.com; angus.mcquibban@utoronto.ca)

S.N., 0000-0001-5765-4695; K.S., 0000-0002-8107-1163; C.M.Y., 0000-0003-4507-556X

# RESULTS

## PGAM5 oligomerization levels play an important role for DRP1 binding

Under different stress conditions, mitochondrial morphology depends on DRP1 function (Sprenger and Langer, 2019; Sabouny and Shutt, 2020). Previous studies have shown that DRP1 activity is dependent on its phosphorylation status and activation of different post-translational modifiers of DRP1 depends on the type of stress condition. During mitochondrial depolarization, DRP1 phosphorylation level relies on the activity of the cytosolic phosphatase calcineurin, whereas, during glucose starvation, it depends on the activity of protein kinase A (Sabouny and Shutt, 2020). As PGAM5 interacts with DRP1 mostly under stress conditions (Lu et al., 2014), we first examined the role of PGAM5 in carbonyl cyanide *m*-chlorophenylhydrazone (CCCP) stress-dependent mitochondrial fission (Fig. S1A–C). Under CCCP-treated conditions, in control U2OS cells, the mitochondrial network undergoes fission resulting in the formation of small punctate fragmented mitochondria; however, in, PGAM5-depleted cells, CCCP treatment led to the significant increase in the formation of ring-shaped (or donut-shaped) mitochondria, which is an indicator of improper fission. These data suggest that PGAM5 plays an important role in CCCP stress-dependent mitochondrial fission.

Previous studies (Zeb et al., 2021; Sugo et al., 2018; Lo and Hannink, 2006) have shown that PGAM5 function is primarily regulated by its stability. PGAM5 is constantly degraded by a Keap1-mediated ubiquitylation pathway, which is inhibited during reactive oxygen species (ROS) induction (Sugo et al., 2018; Wai et al., 2016). We therefore assessed PGAM5 degradation in the presence of CCCP, a widely used protonophore that causes mitochondrial depolarization in U2OS cells (Fig. 1A; Fig. S2). We observed that, with moderate CCCP stress (20 µM for 2 h), there is a marginal but not significant increase in PGAM5 protein level (Fig. 1A, Fig. S2A). In mammalian cells, PGAM5 is expressed as a 32 kDa protein, which is cleaved by the IMM proteases PARL and OMA1 (Wai et al., 2016). In the CCCP-treated cells, we observed that the unprocessed form of PGAM5, PGAM5-L, is completely cleaved into PGAM5-P (cleaved form) (Fig. 1A; Fig. S2B,C). To understand the rate of PGAM5 degradation during CCCP treatment, we further assessed the stability of the individual L or P forms in presence of the translation inhibitor cycloheximide and the proteasomal inhibitor MG132. Under CCCP-treated conditions, MG132 treatment could not further stabilize the protein levels of PGAM5-L or PGAM5-P (Fig. 1A; Fig. S2D,E). Together, these data suggest that PGAM5 function does not rely solely on its stability but might have other modes of regulation.

In mammalian cells, PGAM5 is present in either a dimeric or dodecameric form (Ruiz et al., 2019) and previous studies (Wilkins et al., 2014; Ruiz et al., 2019; Ma et al., 2020; Siebert et al., 2022) have shown that formation of these different PGAM5 oligomers depends on mitochondrial stress. In healthy cells, PGAM5 is in a dodecameric form whereas it changes to a dimeric form under stress (Siebert et al., 2022). In our previous study, we observed that the PGAM5 interaction with MFN2 or DRP1 changes during mitochondrial stress (Nag et al., 2023a). Therefore, we hypothesized that the formation of different PGAM5 oligomers might regulate its binding to its potential substrates. To test this hypothesis, we compared the interaction patterns of wild-type (WT) PGAM5 (capable of oligomerizing into both dodecameric and dimeric forms) (Ruiz et al., 2019) or ΔMM WT PGAM5 (capable of oligomerizing into the dimeric form only) (Ruiz et al., 2019) with MFN2 and DRP1 in the presence and absence of CCCP (Fig. 1B–J). In HEK293 cells, both WT PGAM5 and ΔMM

WT PGAM5 GFP showed binding to MFN2 under steady-state conditions but the binding became significantly weaker under CCCP-treated conditions (Fig. 1C–E). However, under steady-state conditions, ΔMM WT PGAM5 showed reduced affinity towards MFN2 compared to the WT PGAM5 form (Fig. 1C,F). By contrast, both WT PGAM5 and ΔMM WT PGAM5 showed weaker binding to DRP1 under steady-state conditions which significantly increased under the CCCP-treated conditions (Fig. 1G–I). Interestingly, under CCCP-treated conditions, ΔMM WT PGAM5 showed a stronger interaction with DRP1 than the WT PGAM5 (Fig. 1G,J). Although there was some variability in expression levels of the exogenous PGAM5, these data suggest that dodecameric PGAM5 preferably binds to MFN2 whereas dimeric PGAM5 preferentially binds to DRP1.

Next, we aimed to check the functional output of these interactions. To accomplish this, we examined the mitochondrial morphology in U2OS cells overexpressing WT PGAM5 or ΔMM WT PGAM5 under CCCP stress (Fig. 2A–C). Cells overexpressing WT PGAM5 or ΔMM WT PGAM5 showed a higher rate of fragmentation compared to the control cells. Additionally, cells expressing ΔMM WT PGAM5 exhibited a higher rate of mitochondrial fragmentation compared to those expressing WT PGAM5, which suggests an increased activity of DRP1.

In mammalian cells, under steady-state condition, PGAM5 exists primarily as dodecamers, whereas under CCCP-treated conditions, it attains a dimeric form (Siebert et al., 2022). In our experiments, in HEK293 cells, under CCCP-treated conditions, the dimeric form of PGAM5 showed a higher affinity for DRP1. Accordingly, U2OS cells overexpressing the dimeric mutant ΔMM WT PGAM5 showed a higher degree of fragmentation than the WT. To assess the PGAM5 oligomerization in these different cell types, we evaluated the binding of endogenous PGAM5 with the exogenous WT PGAM5 or the ΔMM WT PGAM5 in HEK293 cells (Fig. S3A,B). Under both DMSO- and CCCP-treated conditions, the endogenous PGAM5 showed a higher affinity for WT PGAM5 than the ΔMM WT PGAM5.

We also checked the amount of endogenous PGAM5 oligomers and dimers present in U2OS cells by native-PAGE analysis (Fig. S3C). Under steady-state conditions, PGAM5 mostly formed the higher-order oligomers. Comparatively, under CCCP-treated condition we could also see an increase in the accumulation of dimeric forms. Together, these data suggest that under CCCP-treated condition, PGAM5 assembles into the dimeric conformation, and PGAM5 association and oligomerization states play an important role in regulating the functional output of its interaction with MFN2 and DRP1.

## Cytosolic fractions of PGAM5 play a crucial role in binding to DRP1

Previous literature suggests that PGAM5 is distributed in both the IMM and OMM, as well as in ER–mitochondria contact sites (Lo and Hannink, 2006, 2008; Sekine at al., 2012; Sugo et al., 2018). Furthermore, under stress conditions, a fraction of PGAM5 localizes to the cytosol (Bernkopf et al., 2018). Under steady-state conditions, DRP1 primarily remains cytosolic, but is recruited to the OMM during mitochondrial stress conditions (Yu et al., 2021). From the analysis of immunofluorescence microscopy images, we also noted that the cytosolic distribution of the ΔMM WT PGAM5 was greater than that seen for WT PGAM5 (Fig. S4A,B). From these observations, we hypothesized that PGAM5 localization might also be an important factor in regulating its activity. It is possible that it is cytosolic PGAM5 that is responsible for initiating DRP1

Journal of Cell Science

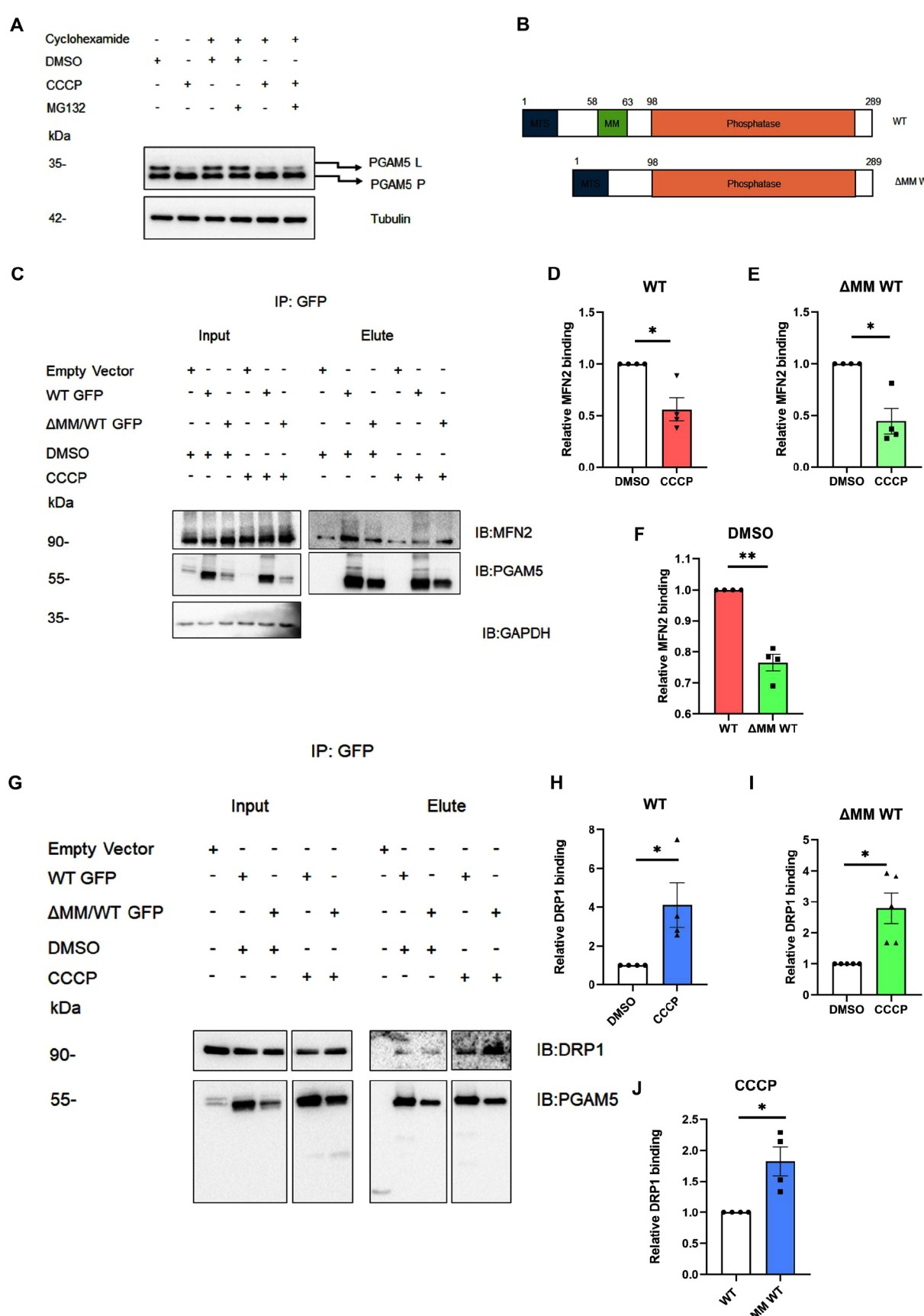

**Fig. 1.** See next page for legend.

**Fig. 1. PGAM5 oligomerization plays important roles in binding its substrates.** (A) Western blot from total U2OS cell lysates showing the PGAM5 protein level and cleavage under different treatment conditions. Representative of four repeats. (B) Schematic representation of the PGAM5 WT and ΔMM WT mutant. (C) Representative western blots showing the interaction pattern between PGAM5 WT GFP or ΔMM WT GFP mutants expressed in HEK293 cells, with MFN2 under DMSO- and CCCP-treated conditions. (D) MFN2 binding was normalized to PGAM5 WT GFP pulldown and the relative binding amount in DMSO- and CCCP-treated conditions were plotted. The bar graph shows the comparison of the relative MFN2–PGAM5 WT binding between DMSO- and CCCP-treated conditions. Mean ±s.e.m. ($n$=4). *$P$=0.0291 (two-tailed paired $t$-test). (E) MFN2 binding was normalized to PGAM5 ΔMM WT GFP pulldown and the relative binding amounts in DMSO- and CCCP-treated conditions were plotted. The bar graph shows the comparison of the relative MFN2–ΔMM WT GFP binding between DMSO and CCCP-treated conditions. Mean±s.e.m. ($n$=4). *$P$=0.0206 (two-tailed paired $t$-test). (F) MFN2 binding was normalized to PGAM5 GFP pulldown and relative binding amount in with WT and ΔMM WT under DMSO conditions were plotted. The bar graph shows the comparison of the relative MFN2 binding between WT GFP and ΔMM WT GFP under DMSO treated conditions. Mean±s.e.m. ($n$=4). **$P$=0.0029 (two-tailed paired $t$-test). (G) Representative western blots showing the interaction pattern between PGAM5 WT GFP or ΔMM WT GFP mutant expressed in HEK293 cells, with DRP1 under DMSO- and CCCP-treated conditions. (H) DRP1 binding was normalized to PGAM5 WT GFP pulldown and the relative binding amounts in the DMSO- and CCCP-treated conditions were plotted. The bar graph shows the comparison of the relative DRP1–WT binding between DMSO- and CCCP-treated conditions. Mean±s.e.m. ($n$=4). *$P$=0.0492 (two-tailed paired $t$-test). (I) DRP1 binding was normalized to PGAM5 ΔMM WT GFP pulldown and the relative binding amounts in the DMSO- and CCCP-treated conditions were plotted. The bar graph shows the comparison of the relative DRP1–ΔMM WT binding between DMSO- and CCCP-treated conditions. Mean±s.e.m. ($n$=4). *$P$=0.0216 (two-tailed paired $t$-test). (J) DRP1 binding was normalized to PGAM5 GFP pulldown and relative binding amount in with WT and ΔMM WT under CCCP treated conditions were plotted. Mean±s.e.m. ($n$=4). The bar graph shows the comparison of the relative DRP1 binding between WT GFP and ΔMM WT GFP under CCCP-treated conditions. *$P$=0.0380 (two-tailed paired $t$-test). IP, immunoprecipitation; IB, immunoblot. Input in C and G shows 5%.

recruitment to mitochondria. To investigate this, we designed a cytosol-localized PGAM5 mutant, Δ35WT PGAM5, lacking the N-terminal 35 amino acids (Fig. 2D,E). By immunofluorescence microscopy analysis in U2OS cells, we confirmed that this mutant primarily localizes to the cytosol (Fig. 2D). Under both steady-state and CCCP-treated conditions, cells overexpressing the Δ35WT PGAM5 mutant showed a higher degree of fragmentation compared to the control (Fig. 2E–H). Interestingly, under steady-state conditions, cells overexpressing Δ35WT showed higher fragmentation than the WT (Fig. 2H). These data suggest that cytosolic PGAM5 activates DRP1-mediated mitochondrial fragmentation.

## PGAM5 cleavage is necessary for binding DRP1 and mitochondrial fragmentation
Previous studies have shown that during CCCP-induced stress, cleaved PGAM5 translocates to the cytosol (Bernkopf et al., 2018; Baba et al., 2021). Therefore, we hypothesized that PGAM5 cleavage is necessary for DRP1 binding and stress-induced mitochondrial fragmentation. To test this hypothesis, we first checked the interaction pattern of endogenous PGAM5 and DRP1 in presence and absence of CCCP stress (Fig. S5A,B). In HEK293 cells, DRP1 WT FLAG showed interaction selectively with the cleaved PGAM5 form (PGAM5-P) and this interaction became stronger in presence of CCCP stress.

Next, we compared the binding between WT or cleavage-resistant forms of PGAM5 (Nag et al., 2023a; Siebert et al., 2022; Sekine et al., 2012) and DRP1 in the presence and absence of CCCP

(Fig. 3A–C). PGAM5 is cleaved at residue serine 24 by the mitochondrial IMM protease PARL and we observed this similar cleavage pattern in PGAM5 WT GFP and endogenous PGAM5 in HEK293 cells (Fig. S6A). A previous literature has shown that mutation of Ser→Phe (S24F) prevents PARL-mediated PGAM5 cleavage (Sekine et al., 2012). PGAM5 is also cleaved by OMA1, an IMM protease, but the cleavage site itself remains unknown (Wai et al., 2016). To overcome this issue, we used an outer membrane-locked PGAM5 mutant (denoted 'OMM mutant') consisting of an N-terminal signal sequence derived from Tom20, an OMM-localized protein, followed by the trans-membrane and phosphatase domains of PGAM5 (Fig. 3A). In HEK293 cells, under CCCP-treated conditions, PGAM5 cleavage-resistant mutants show reduced binding with DRP1 compared to WT PGAM5 (Fig. 3B,C). We also compared the CCCP-induced mitochondrial fragmentation between cells overexpressing either PGAM5 WT, S24F and the OMM mutant in U2OS cells (Fig. 3D–F). Cells overexpressing the PGAM5 cleavage-resistant mutants showed less fragmentation than the cells overexpressing PGAM5 WT, suggesting that cleavage is required for DRP1-mediated mitochondrial fragmentation in the context of exogenous expression of these PGAM5 constructs.

Next, we checked the cytosolic localization of PGAM5 WT and mutants under CCCP stress by mitochondrial fractionation in U2OS cells (Fig. S6B,C). PGAM5 appeared in both crude mitochondrial or mitochondrial-enriched and supernatant or cytosol-enriched fractions. However, under CCCP-treated conditions, PGAM5 WT showed an enhanced presence in cytosol-enriched fractions indicating its increased cytosolic distribution compared to the cleavage-resistant forms. Together, we conclude that the localization and oligomerization or association state of PGAM5 play equally important roles in binding to MFN2 and DRP1.

From these results, we hypothesized a working model (Fig. 3G) where, under steady-state conditions, PGAM5 adopts a dodecameric state and binds MFN2, and DRP1 remains cytosolic (panel 1). In the presence of stress, PGAM5 switches to a dimeric form and dissociates from MFN2 (panel 2). MFN2 is further phosphorylated and ubiquitylated by kinases and ubiquitin ligases and degrades via proteosomal degradation (panel 2). This stress stimulus also initiates PARL-mediated PGAM5 cleavage (panel 3). PARL-cleaved PGAM5 translocates to the cytosol (panel 4). Cytosolic PGAM5 interacts and dephosphorylates DRP1, which gets recruited to the mitochondria and initiates mitochondrial fragmentation (panel 5). As both the dimeric and cleavage-resistant forms of PGAM5 show reduced interaction with MFN2 under stress (Figs 1C and 3B), we conclude that there might be other contributing factors regulating PGAM5–MFN2 interactions.

## PGAM5 enhances mitochondrial connectivity during starvation by regulating DRP1 activity
Mitochondrial dynamics are an indicator of cellular stress and metabolic health. In the presence of depolarization stress, mitochondria undergo fission resulting in a fragmented network. Conversely, during glucose starvation, fission is downregulated, resulting in the formation of a highly connected mitochondrial network. Stress-induced mitochondrial hyperfusion is dependent on the function of several crucial mitochondrial dynamics regulatory proteins like SLP-2, L-OPA1, MFN1 and DRP1 (Gomes et al., 2011; Rambold et al., 2011; Tondera et al., 2009).

To understand the effect of metabolic stress on PGAM5 function, we first checked the mitochondrial morphology of PGAM5-depleted cells during glucose starvation (Fig. 4A–C). In nutrient-rich environments, the mitochondrial network is maintained in a

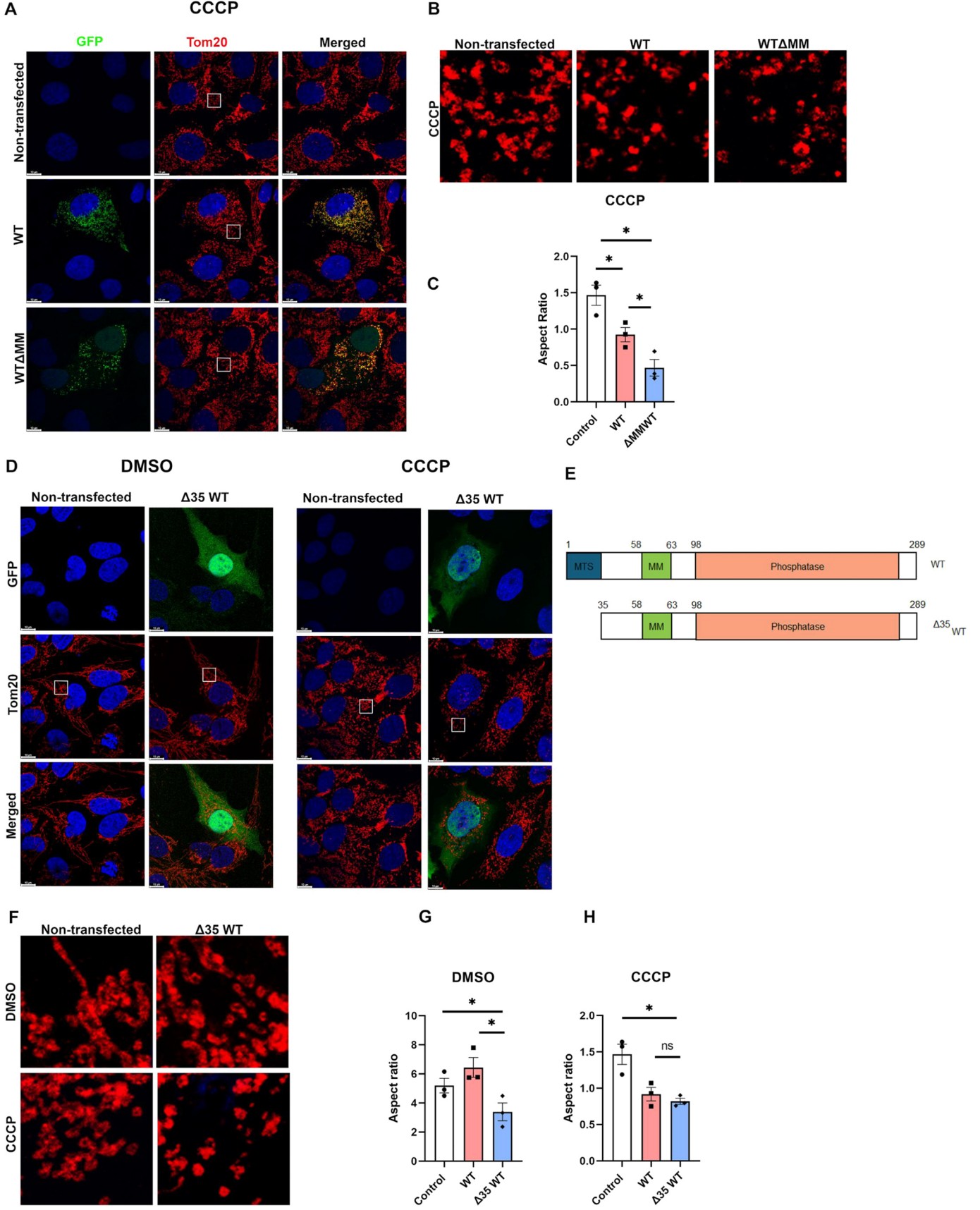

**Fig. 2.** See next page for legend.

**Fig. 2. Change in PGAM5 oligomerization and localization play equally important role in mitochondrial dynamics.** (A) Confocal images of U2OS cells transiently transfected with either PGAM5 WT or ΔMM WT GFP. Scale bars: 10 μm. (B) Magnified views of the overexpression phenotype in the areas highlighted by the white boxes in A. Scale bars: 5 μm. (C) The bar diagram represents the quantification of the mitochondrial aspect ratio. Mean ±s.e.m. (*n*=3) *P*-values (**P*<0.05): control, WT, 0.118; Control, ΔMM WT, 0.0170; WT, ΔMM WT, 0.0262 (two-tailed paired *t*-test). (D) Confocal images of U2OS cells transiently transfected with either PGAM5 WT or Δ35 WT GFP. Scale bars: 10 μm. (E) Schematic representation of PGAM5 WT and Δ35 WT mutant. (F) Magnified views of the overexpression phenotype in the areas highlighted by the white boxes in E. Scale bars: 5 μm. (G) The bar diagram represents the quantification of the mitochondrial aspect ratio under DMSO-treated condition. Mean±s.e.m. (*n*=3) using GraphPad Prism. *P*-values (**P*<0.05): Control, WT, *P*=0.0315; Control, Δ35 WT, 0.0469; WT, Δ35 WT, 0.0106 (two-tailed paired *t*-test). (H) The bar diagram represents the quantification of the mitochondrial aspect ratio under CCCP-treated condition. Mean±s.e.m. (*n*=3) using GraphPad Prism. *P*-values, Control, WT, *P*=0.0117; Control, Δ35 WT, **P*=0.0275; WT, Δ35 WT, 0.2137 (ns, not significant) (two-tailed paired *t*-test). The difference between control and WT is always significant and has not been annotated with an asterisk.

partially fragmented state, whereas after 3 h of starvation with Hank's balanced saline solution (HBSS), control U2OS cells showed a moderate increase in tubular mitochondrial networks. Conversely, during starvation, PGAM5 knockdown (KD) cells showed a partially connected but mostly fragmented mitochondrial network. By contrast, under starvation conditions, PGAM5 WT–GFP-overexpressing cells exhibited a highly interconnected tubular mitochondrial network as compared to control cells (Fig. 4A,B,D). These data suggest that PGAM5 has an important role in the starvation-induced change in mitochondrial dynamics.

During HBSS-dependent glucose starvation, DRP1 plays a major role in shaping the mitochondrial morphology, and its activity depends on its phosphorylation level (Gomes et al., 2011; Rambold et al., 2011). As PGAM5 functions as a DRP1 phosphatase, we assessed the binding of PGAM5 to DRP1 in U2OS cells (Fig. 4E–H). Interestingly, during starvation, PGAM5 WT interacted with DRP1 (Fig. 4E,F) with a higher affinity as compared to the control condition, but not with MFN2 (Fig. 4E,G). Also, PGAM5 WT–GFP showed a reduced affinity towards endogenous PGAM5 under starvation conditions, which suggests a shift to the dimeric state (Fig. 4E,H). Together, these results suggest that during HBSS-induced starvation PGAM5 regulates mitochondrial morphology in a DRP1-dependent manner.

### PGAM5 dephosphorylates DRP1 p616 during HBSS starvation condition

DRP1 has two major phosphorylation sites Ser616 and Ser637. Phosphorylation of Ser616 enhances fission whereas phosphorylation of Ser637 reduces fission (Tilokani et al., 2018). HBSS-induced glucose starvation enhances the phosphorylation level of Ser637 and reduces the phosphorylation level of Ser616 resulting in mitochondrial hyperfusion (Gomes et al., 2011; Rambold et al., 2011). In our HBSS-starved U2OS cells, PGAM5 showed a higher affinity for DRP1 and depletion of PGAM5 results in the perturbation of the mitochondrial network formation. Therefore, we hypothesized that PGAM5 likely regulates DRP1 Ser616 phosphorylation levels during HBSS starvation. To check this hypothesis, we assessed the DRP1 Ser616 and Ser637 phosphorylation level in control and PGAM5 KD U2OS cells (Fig. 5A–D). Similar to what was found in previous literature, HBSS starvation decreased Ser616 phosphorylation and enhanced Ser637 phosphorylation level in control cells (Fig. 5A–C). However, interestingly, under starvation conditions, PGAM5 KD cells showed significantly increased Ser616 phosphorylation but no significant change in the Ser637

phosphorylation level (Fig. 5A,D,E). Therefore, we propose that PGAM5 likely regulates DRP1 Ser616 phosphorylation levels during HBSS starvation and thereby can also regulate mitochondrial dynamics under starvation conditions by inhibiting fusion.

### HBSS starvation stabilizes PGAM5 protein level as well as enhancing its cleavage

To further understand the mechanism of PGAM5 regulation during starvation, we compared PGAM5 protein levels between control and glucose starvation condition in U2OS cells (Fig. 6A,B). Interestingly, during glucose starvation conditions, PGAM5 protein levels were significantly increased compared to control. To understand the rate of PGAM5 cleavage in HBSS starved cells, we checked PGAM5 cleavage in the presence of cyclohexamide and MG132 and found an increase in PGAM5 cleavage (Fig. 6C). In summary, we propose a model (Fig. 6D) where PGAM5 function is regulated by two distinct mechanisms: (1) a change in oligomerization and cleavage, and (2) blockage of degradation. Mitochondrial depolarization stress (CCCP) or metabolic stress (HBSS-induced glucose starvation) induces changes in PGAM5 oligomerization and cleavage resulting in changes in localization (mode 1). Glucose starvation can also prevent PGAM5 degradation thereby increasing its stability (mode 2). Depending on the type of stress stimulus, PGAM5 function is subjected to these two modes of regulation to maintain the mitochondrial homeostasis.

### DISCUSSION
PGAM5 plays a crucial role in various cellular processes, including mitophagy, apoptosis, necrosis and the regulation of mitochondrial dynamics. As the dysregulation of PGAM5 is associated with multiple human diseases, it is important to understand the molecular mechanisms regulating its function. In this study, we have identified an important and previously uncharacterized mechanism by which PGAM5 regulates mitochondrial dynamics. We show that PGAM5 oligomerization or association states play an important role in its interaction with MFN2 and DRP1. During mitochondrial depolarization stress and/or metabolic stress, PGAM5 changes its oligomerization or association state and is cleaved by IMM-localized mitochondrial proteases. A fraction of cleaved PGAM5 then translocates to the cytosol, which initiates DRP1 activation. These observations suggest that, under stress, PGAM5 regulates DRP1 function to induce changes in mitochondrial dynamics.

Previous studies have identified PGAM5 as a regulator of apoptosis. PGAM5-mediated DRP1 activation can trigger programmed necrosis in multiple types of cancer cells (Wang et al., 2012; Zhang et al., 2018). However, unexpectedly, elevated levels of PGAM5 are also associated with different types of cancer (Cheng et al., 2018; Kang et al., 2015; Cheng et al., 2021). Our demonstration of PGAM5 function provides a mechanistic explanation for this previously reported PGAM5 phenotype.

PGAM5 cleavage has been previously shown to be an indicator of cellular stress (Sekine et al., 2012; Wai et al., 2016; Uoselis et al., 2023). Under stress, PGAM5 is cleaved at its N-terminal end by the IMM-resident proteases PARL and OMA1. However, the functional significance of this cleavage was heretofore unknown. Our results now demonstrate that this cleavage enhances translocation of PGAM5 to the cytosol and facilitates PGAM5–DRP1 binding.

In previous studies, PGAM5 has been primarily shown to be an activator of fission (Wang et al., 2012; Sugo et al., 2018). In the presence of ROS or depolarization stress, PGAM5 dephosphorylates DRP1 and activates it, which in turn enhances fission. We similarly observed an interaction between PGAM5 and DRP1. Under CCCP-induced stress, PGAM5 shows a higher affinity towards

Journal of Cell Science

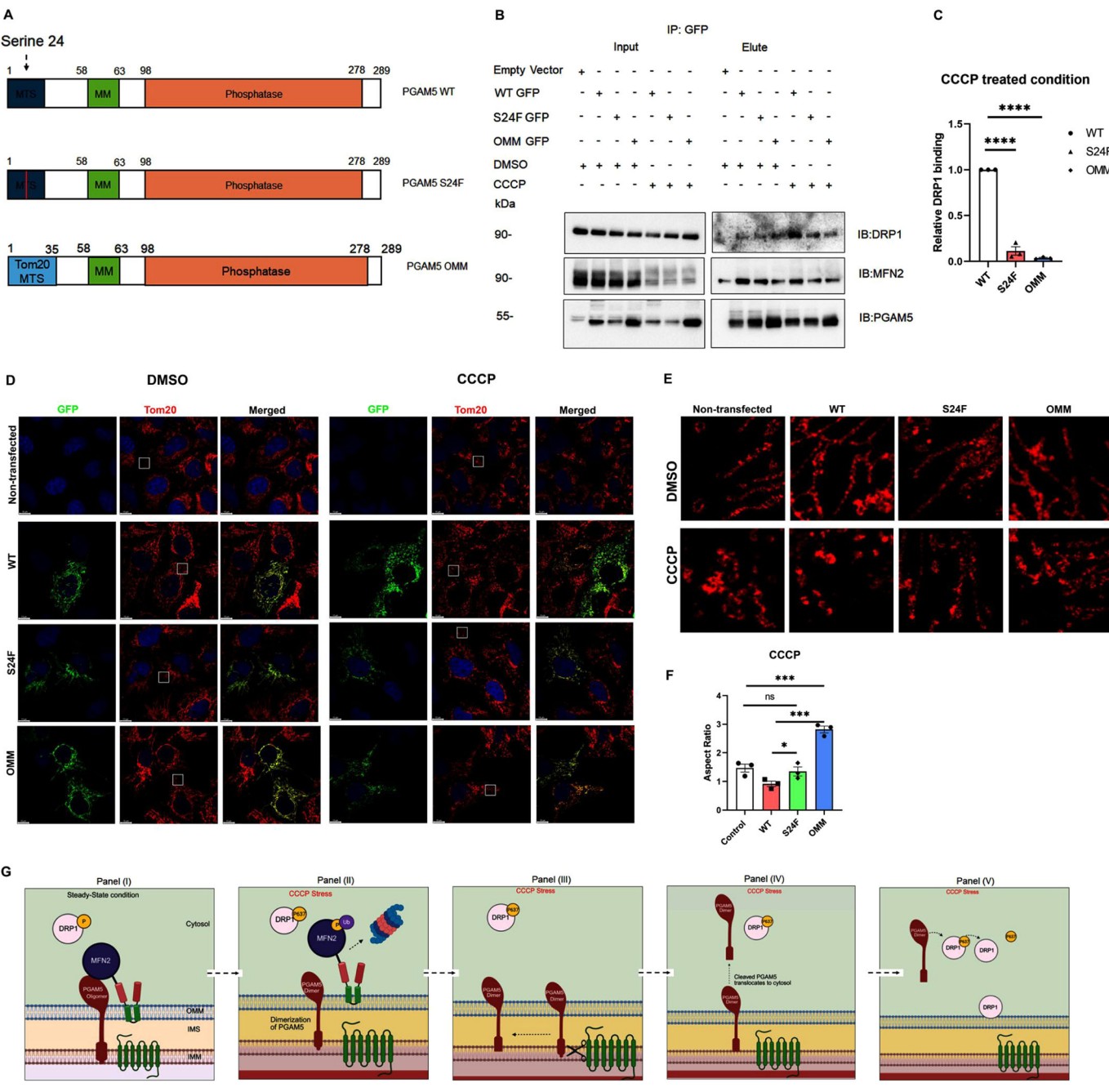

**Fig. 3. PGAM5 cleavage plays an important role in PGAM5–DRP1 interaction.** (A) Schematic representation of PGAM5 WT, and the S24F and OMM mutant. (B) Representative western blots showing the interaction pattern between PGAM5 WT or S24F or OMM GFP mutants expressed in HEK293 cells with MFN2 and DRP1 under DMSO- and CCCP-treated conditions. (C) DRP1 binding was normalized to PGAM5 GFP pulldown and the relative binding amounts for the WT, S24F and OMM mutant under CCCP-treated conditions were plotted. Mean±s.e.m. ($n$=4). ****$P$<0.0001 (WT, S24F and WT, OMM; one-way ANOVA with Dunnett's multiple comparison post test). (D) Confocal images of U2OS cells transiently transfected with either PGAM5 WT or S24F or OMM GFP. Scale bars: 10 μm. (E) Magnified views of the overexpression phenotype in the areas highlighted by the white boxes in D. Scale bars: 5 μm. (F) The bar diagram represents the quantification of the mitochondrial aspect ratio under the CCCP-treated condition. Mean±s.e.m. ($n$=3). $P$-values (*$P$<0.05; ***$P$<0.001; ns, not significant): Control, WT, 0.0117; Control, S24F, 0.0862 (not significant); Control, OMM, 0.0003; WT, S24F, 0.0285; WT, OMM, 0.0004 (two-tailed paired $t$-test). (G) Working model of PGAM5-dependent regulation of mitochondrial dynamics under CCCP-treated conditions. Created in BioRender by Nag, S., 2025. From right to left: https://BioRender.com/rrxhy8t, https://BioRender.com/63ovtgg, https://BioRender.com/pfzah6u, https://BioRender.com/a26k150, https://BioRender.com/8m919kp. These figures were sublicensed under CC-BY 4.0 terms.

DRP1 and depletion of PGAM5 results in improper fission. Additionally, our results show that, under metabolic stress (HBSS-induced glucose starvation), PGAM5 also can regulate DRP1 phosphorylation and function leading to mitochondrial fusion. Overall, our study shows that PGAM5 function is regulated by two modes, which further depends on the cellular stress stimulus.

PGAM5 has emerged as a clinically significant mitochondrial phosphatase due to its multifaceted role in regulating mitochondrial dynamics, cell death pathways and stress responses. Owing to its involvement in modulating mitophagy, apoptosis and necroptosis pathways, PGAM5 functions as a key player in the pathogenesis of various diseases, including neurodegenerative disorders and cancer.

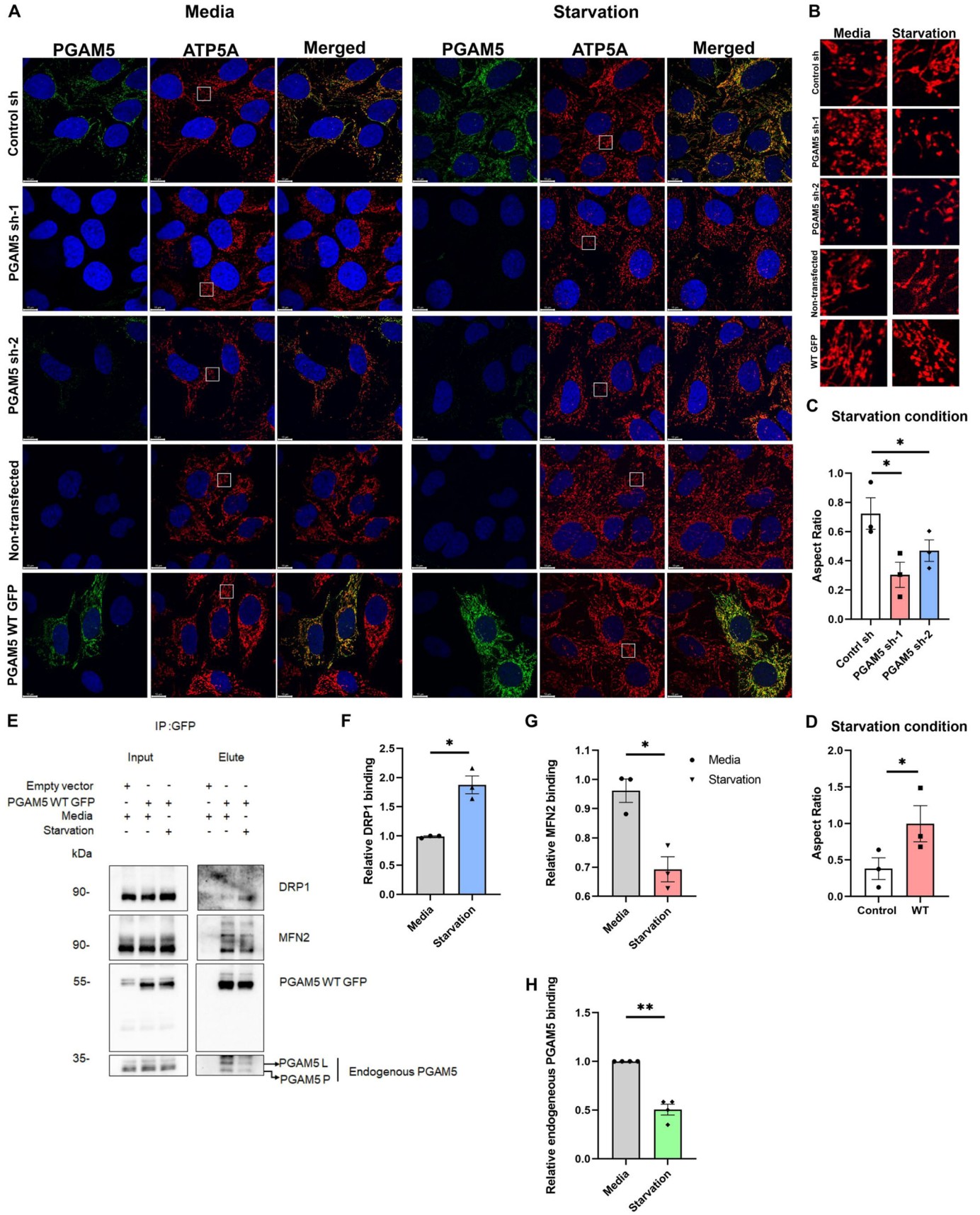

**Fig. 4.** See next page for legend.

**Fig. 4. PGAM5 enhances mitochondrial connectivity during starvation conditions by regulating DRP1 function.** (A) Confocal images of U2OS cells stably expressing control shRNA (control sh) or shRNA against PGAM5 (PGAM5 sh-1 or sh-2) (top three panels) when cultured with medium (media) and under HBSS starvation conditions. Confocal images of U2OS cells transiently transfected with PGAM5 WT or non-transfected control under when cultured with medium (media) and under HBSS starvation conditions (fourth and fifth panel). Scale bars: 10 μm. (B) Magnified views of the overexpression phenotype in the areas highlighted by the white boxes in A. Scale bars: 5 μm. (C) The bar diagram shows the aspect ratio in control and PGAM5 KD cells under starvation condition. Mean±s.e.m. (*n*=3). *P*-values (\**P*<0.05): control sh, PGAM5 sh-1, 0.0128; control sh, PGAM5 sh-2, 0.0312 (two-tailed paired *t*-test). (D) The bar diagram shows the aspect ratio in control and PGAM5 WT overexpressing cells under starvation condition. Mean±s.e.m. (*n*=3). *P*-values: control, WT, 0.0353 (two-tailed paired *t*-test). (E) Representative western blots showing the interaction pattern between PGAM5 WT GFP with MFN2, DRP1 and endogenous PGAM5 in U2OS cells when cultured with medium (media) and under HBSS starvation conditions. IP, immunoprecipitation. Input shows 5%. (F) DRP1 binding was normalized to PGAM5 WT GFP pulldown and relative binding amounts when cultured with medium (media) and under HBSS-induced starvation conditions were plotted. The bar graph shows the comparison of the relative DRP1 binding between media and HBSS-induced starvation conditions. Mean±s.e.m. (*n*=3). \**P*=0.0278 (two-tailed paired *t*-test). (G) MFN2 binding was normalized to PGAM5 WT GFP pulldown and relative binding amounts when cultured with medium (media) and under HBSS-induced starvation conditions were plotted. The bar graph shows the comparison of the relative MFN2 binding between media and HBSS-induced starvation conditions. Means±s.e.m. (*n*=3). \**P*=0.0375 (two-tailed paired *t*-test). (H) Endogenous PGAM5 binding was normalized to PGAM5 WT GFP pulldown and relative binding amounts when cultured with medium (media) and under HBSS-induced starvation conditions were plotted. The bar graph shows the comparison of the relative endogenous PGAM5 binding between media and HBSS-induced starvation conditions. Means±s.e.m. (*n*=3). \*\**P*=0.0030 (two-tailed paired *t*-test).

In neurodegenerative diseases, such as Parkinson's disease, aberrant PGAM5 activity has been linked to impaired mitochondrial quality control and increased neuronal vulnerability. Conversely, in cancer, PGAM5 is often upregulated, promoting cell survival and metabolic reprogramming under stress conditions, which might contribute to tumor progression and resistance to therapy. Given its dual role in cell survival and death, PGAM5 is being investigated as a potential therapeutic target, where context-specific modulation could help restore cellular homeostasis in disease states. Revealing these mechanisms of PGAM5 regulation will enhance the deeper understanding of PGAM5 function and its disease etiology.

## MATERIALS AND METHODS

### Cell culture and transfection

U2OS cells (obtained from Dr Peter Kim, The Hospital for Sick Children, Toronto, Canada) were cultured in McCoy's 5A (modified) medium (Gibco, 16600082) supplemented with 10% fetal bovine serum (Gibco, 26140079) and supplemented with penicillin-streptomycin (Sigma, P4333) at 37°C in humidified air containing 5% $CO_2$. HEK293T cells (obtained from Dr Peter Kim) were cultured in DMEM (high glucose) medium (Sigma, SLM 241) with 10% fetal bovine serum (Gibco, 16600082) and supplemented with penicillin-streptomycin (Sigma, P4333) at 37°C in humidified air containing 5% $CO_2$. Cells were tested for mycoplasma contamination regularly (e-Myco Mycoplasma PCR detection kit, FroggaBio, 25235). All experiments were performed with low passage cells (up to a maximum of passage 20). 2 μg/ml puromycin (Sigma, P7255) was used as a selection marker for the generation of knockdown-stable cell lines. Cells were transfected using Lipofectamine 2000 (Invitrogen, 11668019) as per the manufacturer's protocol.

### Immunofluorescence

For immunofluorescence microscopy-based experiments, 1 μg of plasmid DNA was used per well of a six-well dish. After 48 h of transfection, cells on coverslips (Thermo Fisher Scientific, 12541B) were washed with 1× PBS, and then fixed using 4% PFA (Thermo Fisher Scientific, 28906). Post-fixation cell permeabilization was done with 0.1% Triton X-100 for 20 min. Next, they were washed with 1× PBS and incubated in 10% goat serum (Gibco, 16210072) for 1 h at room temperature. Coverslips were then incubated with the primary antibodies in PBS for 2 h. Following a PBS wash, coverslips were incubated with Alexa Fluor-conjugated secondary antibodies (Life Technologies) for 1 h, washed with PBS, and mounted onto slides using Fluoromount-G (Invitrogen, 00495802). Images were acquired using a Leica SP8 microscope with a 63× magnification objective using the respective laser channels and Las X software.

### Antibodies and reagents

Antibodies used were against: Tom20 (Santa Cruz Biotechnology, 17764), PGAM5 (Abcam, 126534), ATP5A (Abcam, 14748), MFN2 (Abcam, 205236), DRP1 (sc-271583), GFP (Roche/Sigma, 11814460001), tubulin (sc-53646), GAPDH (sc-47724), DRP1 p616 (Abcam-EPR27387-57), DRP1 p637 (Abcam, 193216), VDAC1 (Abcam, 240128), rabbit IgG conjugated to HRP (Jackson ImmunoResearch Laboratories, 111-035-003), mouse IgG conjugated to HRP (Jackson ImmunoResearch Laboratories, 115-035-003). All antibodies were used at 1:1000. Reagent used were: CCCP (Sigma, C2759), GFP–Agarose beads (Chromotek-GFP-Trap agarose, Proteintech), M2 Flag beads (Sigma, A2220), 4–20% gradient gels (Bio-Rad, 4561094), HBSS (Gibco, 14179075), Protein A beads (Sigma, 16-125), PVDF membrane (Immobilon, IPVH00010), cyclohexamide (Millipore-Sigma C4859), MG132 (Millipore-Sigma, M-8699),Tween-20 (BioShop, 1M23298), skim milk (BioShop, SKI400.500), ECL (BioRad, 11705062), acrylamide solution (Bioshop-ACR010), ammonium persulfate (Bio-Rad, 1610700) and TEMED (Bioshop, 110-18-9).

### shRNAs and plasmids

PGAM5 shRNA-1 was 5′-CCGGCGCCATAGAGACCACCGATATCTC-GAGATATCGGTGGTCTCTATGGCGTTTTTG-3′; PGAM5 shRNA-2 was 5′-CCGGGCCGGAAGCTGTGCAGTATTACTCGAGTAATACTG-CACAGCTTCCGGCTTTTTG-3′; PGAM5L and Tom20 plasmids were a generously gift from Dr Mark Hannink (University of Missouri, Columbia, USA), and Dr Peter Kim, respectively, and further sub-cloned into different vectors. GFP-tagged plasmids are cloned into the pEGFPN1 vector (Clontech). DRP1 WT was Addgene #49152 and later subcloned into V1-Flag vector (Clontech).

### Immunoprecipitation and western blotting

Immunoprecipitation of GFP-tagged proteins was performed using GFP-trap beads (Chromotek, gtak20) as per the manufacturer's protocol. For immunoprecipitation of GFP proteins, HEK293T or U2OS cells were transfected with the required plasmids (4 μg/10 cm dish) by the Lipofectamine transfection method. After 48 h of transfection, cells were treated with indicated stressor for the defined time points (2 h for 20 μM CCCP or 3 h with HBSS) and then lysed in the lysis buffer containing 20 mM Tris-HCl pH 7.4, 200 mM Nacl, 5 mM EDTA, 10 mM $MgCl_2$, 1% Triton X-100, 1× protease inhibitor cocktail (Roche, 4693159001), 1× phosphatase inhibitor (Thermo Fisher, Scientific, 78240) for 30 min on ice. Cell lysates were then pre-cleared using the Protein-A agarose beads (Sigma, 16-125) for 2 h (in the cold room). 5% of the pre-cleared lysates were collected as input and the remaining lysate was incubated with the GFP-trap beads for 6–8 h. Beads were washed four times and eluted in 2× sample buffer by boiling at 90°C for 5 min. Western blots were blocked with 5% skimmed milk before addition of antibodies, then were imaged on the Bio-Rad Chemidoc XRS⁺ machine and further quantified using Image Lab software version 6 (BioRad) by densitometry analysis. Uncropped images of western blots from this paper are shown in Fig. S7.

### Native PAGE analysis

Cells were lysed in mild lysis buffer (50 mM Tris-HCl pH 7.4, 150 mM NaCl, 0.5% Triton X-100 and protease inhibitor cocktail). After centrifugation at 12,000 *g* for 10 min, supernatant proteins were quantified and diluted with 2× native PAGE sample buffer (125 mM Tris-HCl pH 6.8, 30% glycerol and

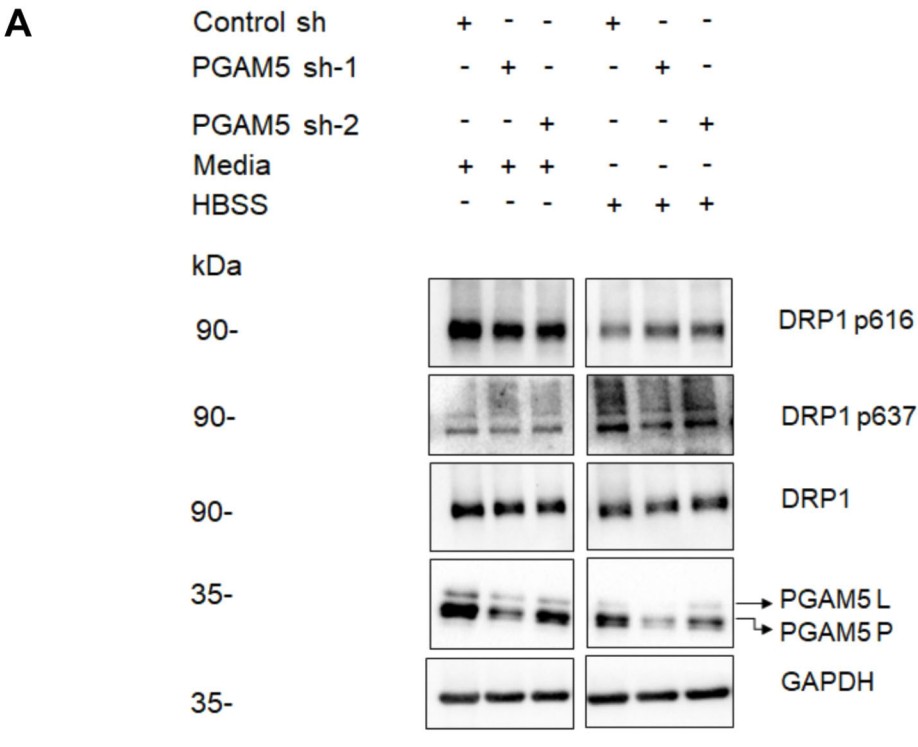

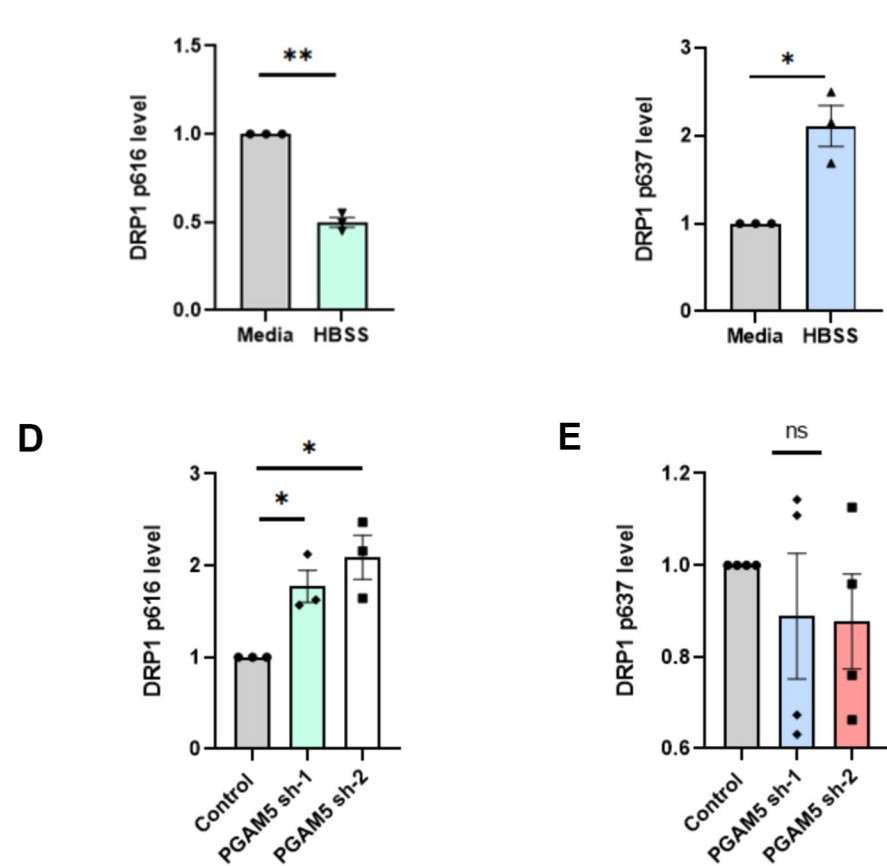

**Fig. 5. PGAM5 dephosphorylates DRP1 phosphorylated at serine 616 during HBSS starvation condition.** (A) Representative western blots showing DRP1 phosphorylation levels [at serine 616 (p616) or serine 637 (p637)] in U2OS cells when cultured with medium (media) and under HBSS-induced starvation conditions in control and PGAM5 KD cells. (B) The bar diagram shows the comparison of the p616 level in control cells when cultured with medium (media) and under starvation conditions. means±s.e.m. (*n*=3). **P=0.0034 (two-tailed paired *t*-test). (C) The bar diagram shows the comparison of DRP1 p637 level in control cells when cultured with medium (media) and under starvation condition. Mean±s.e.m. (*n*=3). *P=0.0417 (two-tailed paired *t*-test). (D) The bar diagram shows the comparison of DRP1 p616 level in control cells and PGAM5 KD cells under starvation condition. Mean±s.e.m. (*n*=3). *P*-values (*P<0.05): control sh, PGAM5 sh-1, 0.0439; control sh, PGAM5 sh-1, 0.0100 (one-way ANOVA with Tukey's multiple comparison *t*-test). (E) The bar diagram shows the comparison of DRP1 p637 level in control cells and PGAM5 KD cells under starvation condition. Means±s.e.m. (*n*=3). *P*-values (ns, not significant): control sh, PGAM5, sh-1, 0.7164; control sh, PGAM5 sh-1, 0.6656 (one-way ANOVA with Tukey's multiple comparison *t*-test).

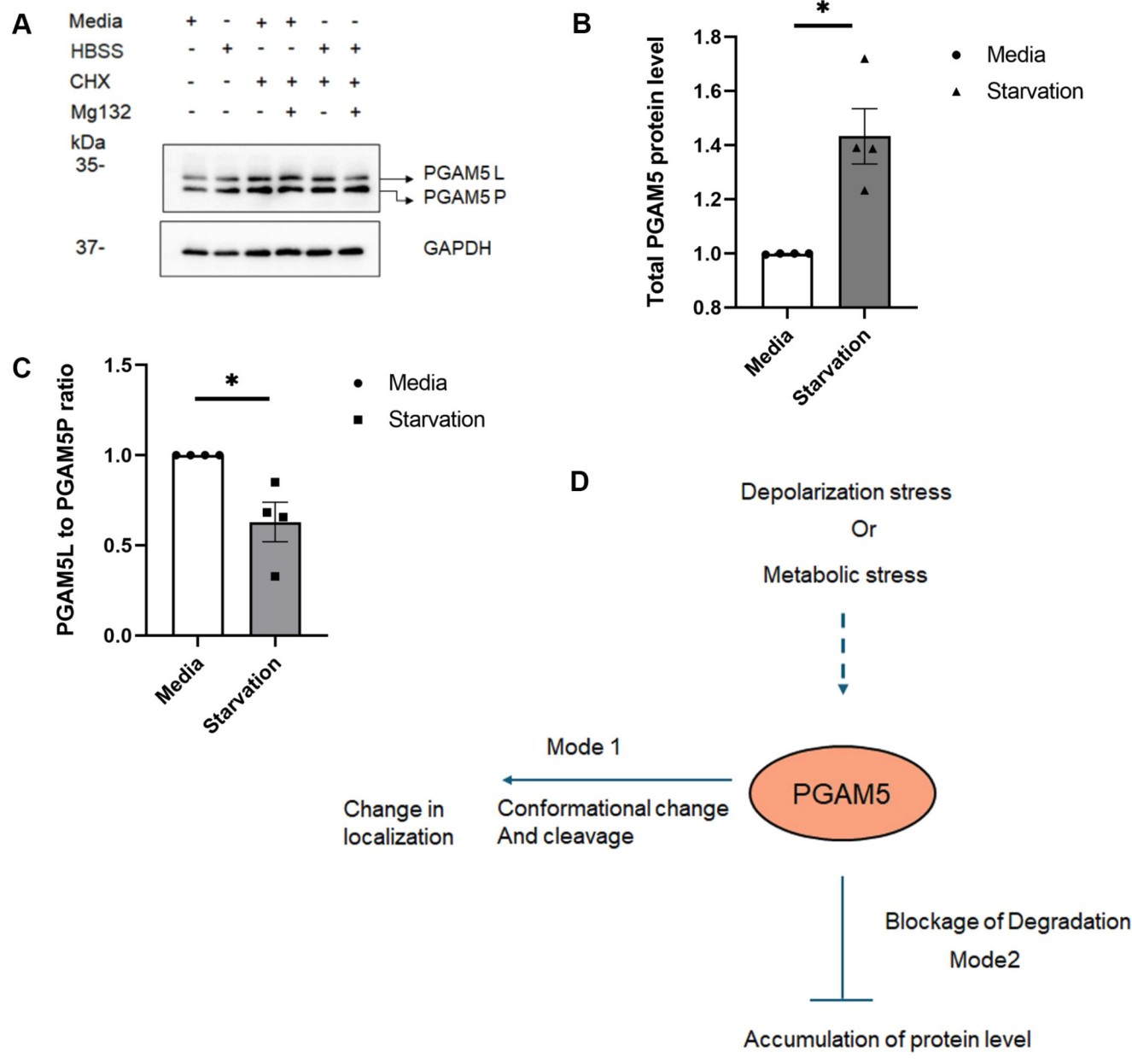

**Fig. 6. HBSS starvation stabilizes PGAM5 protein level as well as enhances its cleavage.** (A) Representative western blots from total cell lysates from U2OS cells showing the PGAM5 protein level and cleavage under when cultured with medium (media) and under HBSS starvation conditions. (B) The bar diagram shows the comparison of normalized values of total PGAM5 protein levels when cultured with medium (media) and under starvation conditions. Means±s.e.m. ($n$=4). *$P$=0.0247 (two-tailed paired $t$-test). (C) The bar diagram shows the comparison of normalized values of PGAM5L and PGAM5P ratios when cultured with medium (media) and under starvation conditions. Mean±s.e.m. ($n$=4). *$P$=0.0427 (two-tailed paired $t$-test). (D) Schematic representation of modes of PGAM5 regulation.

0.1% Bromophenol Blue). 40 μg protein was used for native-PAGE using gels made without SDS. Gels were run with 25 mM Tris0HCL pH 8.4, 192 mM glycine. After electrophoresis, proteins were transferred to PVDF membranes for immunoblotting.

### Image quantification pipeline
Image processing was done through Fiji software as the per the protocol described previously (Nag et al., 2023b; Star protocol).

### Analysis of aspect ratio
All processing was performed using Fiji (Schindelin et al., 2012). Background subtraction was applied using a rolling ball radius of 50 pixels, followed by a median filter with a 1-pixel radius. Images were then binarized using the default thresholding settings. Mitochondria were classified as either

networks or puncta using the trainable Weka Segmentation plugin (Arganda-Carreras et al., 2017), with default settings and the Fast Random Forest classifier (Belgiu and Dragut, 2016). The Analyze Particles tool in Fiji was then used to quantify the area of each mitochondrial category within a given field of view. These values were divided by the total mitochondrial area to yield the percentage of area occupied by networks or puncta. The mitochondrial aspect ratio was then calculated as the ratio of the density (percentage of total area) of mitochondrial networks to that of mitochondrial puncta. Three trials of every experiment have been performed ($n$=3). For every experiment, at least 60–100 cells were analyzed.

### Quantification of rings
Images were first denoised using the 'Denoise and Clarify' functions in Nikon's NIS-Elements software, then binarized in Fiji using Huang

thresholding. Regions of interest (ROIs) containing ring mitochondrial structures were manually identified and isolated using the ROI manager. The area of these structures was quantified using the Analyze Particles tool in Fiji. Ring structure area was divided by the total mitochondrial area to determine the percentage of area occupied by ring structures.

## PGAM5 cleavage and degradation assay

U2OS cells were pre-treated with 100 µM cycloheximide for 30 min as a pre-treatment and during the incubation with DMSO, CCCP or HBSS. 20 µM MG132 were used alongside the incubation to stop the proteasomal degradation of PGAM5.

## Mitochondrial fractionation

U2OS cells were transiently transfected with PGAM5 WT or the other cleavage resistant mutants. After 48 h of incubation, cells were harvested, and the mitochondrial and cytosolic fraction was isolated using the Abcam mitochondria isolation kit (ab110171) as per manufacturer's protocol.

## Statistics

The results were expressed as mean±s.e.m. and analyzed by means of the statistical tests stated in the figure legends. The *P*-values are indicated by asterisks in the figures with the *P*-values. At least three independent biological replicates were performed for all experiments. $P<0.05$ was considered significant. GraphPad Prism version 9 was used to make the graphs and for the evaluation of statistical significance. To calculate the binding affinity of various proteins in co-immunoprecipitations experiments, the amount of eluted proteins were normalized to the bait proteins from the same blot and relative binding amounts were plotted.

### Competing interests

The authors declare no competing or financial interests.

### Author contributions

Conceptualization: S.N.; Formal analysis: K.S., C.M.Y.; Funding acquisition: G.A.M.; Investigation: S.N., K.S., C.M.Y.; Methodology: S.N., K.S., C.M.Y.; Project administration: S.N., G.A.M.; Supervision: C.M.Y., G.A.M.; Validation: S.N.; Visualization: S.N.; Writing – original draft: S.N.; Writing – review & editing: C.M.Y., G.A.M.

### Funding

This work was supported by a Natural Sciences and Engineering Research Council of Canada (NSERC) operating grant. S.N. is supported by the NSERC. Open Access funding provided by University of Toronto. Deposited in PMC for immediate release.

### Data and resource availability

All relevant data and details of resources can be found within the article and its supplementary information.

### First Person

This article has an associated First Person interview with the first author of the paper.

### Peer review history

The peer review history is available online at https://journals.biologists.com/jcs/lookup/doi/10.1242/jcs.263903.reviewer-comments.pdf

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
