## [Peer Review File · Journal of Cell Science]

PGAM5 cleavage and oligomerization equilibrates mitochondrial dynamics under stress by regulating DRP1 function

Sudeshna Nag, Kaitlin Szederkenyi, Christopher M. Yip and Angus McQuibban
DOI: 10.1242/jcs.263903

Editor: Ana Garcia Saez

Review timeline

Original submission:	6 February 2025
Editorial decision:	17 March 2025
First revision received:	17 July 2025
Editorial decision:	14 August 2025
Second revision received:	8 September 2025
Editorial decision:	12 September 2025
Third revision received:	15 September 2025
Accepted:	23 September 2025

Original submission

First decision letter

MS ID#: jcs.263903

MS TITLE: PGAM5 cleavage and oligomerization equilibrates mitochondrial dynamics under stress by regulating DRP1 function

AUTHORS: Sudeshna Nag; Kaitlin Szederkenyi; Christopher M. Yip; Angus McQuibban

ARTICLE TYPE: Research Article

Dear Dr Mcquibban,

We have now reached a decision on the above manuscript.

To see the reviewers' reports and a copy of this decision letter, please go to:

Reviewer 1

Advance summary and potential significance to field

Summary:

The manuscript by Das et al, entitled "PGAM5 cleavage and oligomerization equilibrates mitochondrial dynamics under stress by regulating DRP1 function" examines the role of PGAM5 as a mediator of mitochondrial dynamics. While their intriguing findings suggest a model where both cleavage and oligomerization of PGAM5 regulate this important phosphatase, the manuscript is not yet ready for publication, as several issues still need to be addressed.

Suggestions to Authors:

-The introduction should discuss Drp1 phosphorylation, specifically the 616 and 637 sites and how they promote/inhibit fission.

-The authors mention ring/donut shaped mitochondria and show some representative images. However, there needs to be more quantification of these structures to see if there truly are differences between conditions. It appears that some of these structures are present with PGAM5 shRNA in DMSO treated, as well as in control shRNA cells treated with CCCP. It is also not clear to me if these are just swollen mitochondria, or if they are donut structures described elsewhere that occur when two ends of the same mitochondria fuse together. Scale bars in the zoomed-in panels would be helpful.

-It would be good to indicate in the main text what cell type is being used for the different experiments. They start looking at U2OS cells (Fig 1A) and then switch to HEK293 cells (Fig 1C) and go back and forth. It is not always clear in the main text, which cell type is being used.

-For the HEK293 work, where GFP-tagged PGAM5 is being overexpressed, it is not always clear how much the protein constructs are overexpressed relative to endogenous PGAM5. Some blots show the endogenous PGAM5, others do not. For the IP blots, it would be good to clearly indicate which bands correspond to PGAM5-GFP (full and cleaved) and endogenous PGAM5 (full and cleaved). I noticed some asterisks in Sup Fig 3A, but they were not mentioned in the figure legend. Moreover, the overexpression of the PGAM variants is not always equal (see input lanes for IP - Fig 1C/3B), which could affect the interpretation of some of the findings (i.e., is this factored into calculations for the relative binding...).

-I also wonder about the ability of the PGAM5-GFP to undergo cleavage. Presumably the GFP is at the C-terminus, but a doublet is not always present/evident in the western blots for the PGAM5-GFP, as is seen for the endogenous PGAM5. Similarly, why do the PGAM5-cleavage resistant mutants appear to run at the same size as the WT-PGAM5 (Fig 3B)? Does this mean that the WT-PGAM5 is not cleaved?

-While the WT PGAM5 can form dimer and dodecamer forms, and the MM PGAM5 can only form dimers, I'm not sure that they can conclude the preference of these forms to bind MFN2/DRP1 based on their data. Are there other explanations? Perhaps the MM deletion impacts binding.

-I have some concerns about the quantification of the mitochondrial morphology. The authors use aspect ratio, which is typically a comparison of the length:width ratio of any particular mitochondrial fragment (where the length is defined as the longest axis and the width as the shortest axis). However, the values for the aspect ratio seem to vary between ~0.5 to 3. How do you get a ratio below 1 - wouldn't that mean the length is no longer the longest axis and there is something wrong with how the value is calculated. Conversely, if length is the longest axis, how do you get a ratio above 1? Would average mitochondrial length be more appropriate in the context of looking at mitochondrial fragmentation. It is also not clear from the methods and figure legends what kind of replicates were used. The methods state that 60-100 cells were analyzed, but in the figure legends it says n=3. Does n=3 refer to 3 mitochondria, 3 cells or 3 replicates of 60-100 cells. Are the replicates technical or biological.

-For the mitochondrial fractionation (Sup Fig 3 C/D) there is nothing to indicate how clean the fractions are. It is also not clear why actin was used as a load control for the mitochondrial fraction. Without the proper controls, conclusions from this experiment are meaningless. The authors argue PGAM5 WT was more cytosolic compared to cleavage resistant forms, which may be true. However, if you look at the cytosolic fraction of PGAM5 WT in DMSO -treated cells, there does not appear to be any difference in compared to CCCP treatment. Wouldn't the model predict there should be more cytosolic PGAM5 WT with CCCP?

-The quantification of the starvation induced effects on mitochondrial morphology (Fig 4A/B) appears to be incomplete. Panels C/D only shows the quantification during starvation, and not in normal media. The authors should also show the morphology under normal media, so that we can see if the treatment is inducing mitochondrial hyperfusion as expected - the representative images don't necessarily show this is the case (e.g. mitos in non-transfected cells look the same in normal media and under starvation).

-Authors make a hypothesis about the phosphorylation status of Drp1. This should be very easy to check in their cell lines, as there are commercially available antibodies they are very commonly used. Adding this data would greatly strengthen the conclusions of their model.

Minor Issues:

-The model in Fig 3G needs work. In Panel 1, PGAM5 is shown as a dimer, not a dodecamer. In Panel 2, PGAM5 is just as close to MFN2 as in Panel 1 (should be more distant if it dissociates). It is also not clear why the proteasome is depicted in Panel 2, this is not discussed anywhere in the text.

-The term 'stress' is used throughout the manuscript. However, given that different stresses (CCCP vs Starvation) seem to have distinct effects on PGAM5, the authors should specify which stress they are referring to at all times.

-I'm not sure I'd agree that starvation-dependent morphology is mostly dependent on DRP1 activity (line 168). See <https://pubmed.ncbi.nlm.nih.gov/19360003/>.

-Sup Fig 4A is labelled a MFN2 binding, but in the main text is referred to as DRP1 binding.

-Graph axis in Sup Fig 4B is a bit misleading, be sure to clearly indicate the axis break.

Reviewer 2

Advance summary and potential significance to field

Nag et al present an interesting study regarding the role of the mitochondrial phosphatase PGAM5. They propose that metabolic stress leads to changes in the oligomeric state of PGAM5, reducing its affinity for MFN2, and instead leading it to preferentially bind and dephosphorylate Drp1, leading to Drp1 activation and mitochondrial fission. They also propose that stress-induced cleavage of PGAM5 releases it into the cytosol, where it is capable of binding and activating Drp1. The data presented are interesting, well performed and generally convincing, and will be of great interest to the field.

Comments for the author

While I believe this study is well-performed, I feel several aspects of the model rely too heavily on inferences from previous papers, or on assumptions. For example, the authors do not directly demonstrate that stress changes the oligomeric state of PGAM5, or directly demonstrate preferential binding of MFN2 or Drp1 to either of these forms. They also make the assumption that PGAM5 controls Drp1 activity through dephosphorylation of Drp1 at S616 - however, at no point do they monitor Drp1 phosphorylation in their assays. A key finding from their paper is that cleaved PGAM5 binds Drp1 in the cytosol. If this is the case, then Drp1 should preferentially interact with the cleaved (smaller) form of PGAM5. The authors could potentially perform IPs of Drp1 and examine whether it preferentially immunoprecipitates cleaved PGAM5.

A question that also springs to mind is which of these two mechanisms of action (changed oligomeric state versus cleavage) are most important for Drp1 activation. Since Drp1 appears to bind poorly to mitochondrially-localised PGAM5, is the switch in oligomeric state only a minor contributor to PGAM5's regulation of Drp1 activity, with the main role, at least under stress conditions, being through release of the cleaved fragment?

Minor point - It would be helpful to have more detail about how the mitochondrial morphology analysis was performed. How is the 'Aspect ratio' determined?

First revision

Author response to reviewers' comments**Reviewer 1:**

Suggestions to Authors:

1. The introduction should discuss Drp1 phosphorylation, specifically the 616 and 637 sites and how they promote/inhibit fission.

-This information has been added in the introduction.

2. The authors mention ring/donut shaped mitochondria and show some representative images. However, there needs to be more quantification of these structures to see if there truly are differences between conditions. It appears that some of these structures are present with PGAM5 shRNA in DMSO treated, as well as in control shRNA cells treated with CCCP. It is also not clear to me if these are just swollen mitochondria, or if they are donut structures described elsewhere that occur when two ends of the same mitochondria fuse together. Scale bars in the zoomed-in panels would be helpful.

-The quantification has been added in Supplementary figure 1D. Accordingly, the text, figure legend and the quantification protocol (of the ring structures) has been updated. Scale bar information has been added to the figure legends.

Consistent with our visual observation under the microscope, we see the mitochondria in PGAM5 KD cells as donut-shaped not punctate. We do see some of these structures in the DMSO treated conditions in PGAM5 KD cells also, but the numbers are not statistically significant.

3. It would be good to indicate in the main text what cell type is being used for the different experiments. They start looking at U2OS cells (Fig 1A) and then switch to HEK293 cells (Fig 1C) and go back and forth. It is not always clear in the main text, which cell type is being used.

-The information in the text has been updated. HEK 293 cells were only used for most of the co-IP experiments. U2OS cells have been used for all other experiments.

4. For the HEK293 work, where GFP-tagged PGAM5 is being overexpressed, it is not always clear how much the protein constructs are overexpressed relative to endogenous PGAM5. Some blots show the endogenous PGAM5, others do not. For the IP blots, it would be good to clearly indicate which bands correspond to PGAM5-GFP (full and cleaved) and endogenous PGAM5 (full and cleaved). I noticed some asterisks in Sup Fig 3A, but they were not mentioned in the figure legend. Moreover, the overexpression of the PGAM variants is not always equal (see input lanes for IP - Fig 1C/3B), which could affect the interpretation of some of the findings (i.e., is this factored into calculations for the relative binding...).

-For co-IP experiments 4ug plasmids have been used /10cm dish. This information has been added in the materials and methods section.

IP samples were resolved using 10% SDS-PAGE. In these gels, full length and cleaved PGAM5 GFP bands could not be resolved clearly. The best resolution of PGAM5 GFP (full length and cleaved bands) can be visible only by 4-20% gradient gels. Endogenous PGAM5 bands, full length and the cleaved bands can be best resolved by using 12% SDS-gels. So, in order to avoid confusion only the best resolved gels have been used in the representative images.

The full length and cleaved bands have been marked in the IP blots and the figure legends have been updated.

In the bar diagrams the relative binding is represented which shows the normalized binding based on the GFP protein levels. This information has been also added in the figure legends and materials and methods section.

5. I also wonder about the ability of the PGAM5-GFP to undergo cleavage. Presumably the GFP is at the C-terminus, but a doublet is not always present/evident in the western blots for the PGAM5-GFP, as is seen for the endogenous PGAM5. Similarly, why do the PGAM5-cleavage resistant mutants

appear to run at the same size as the WT-PGAM5 (Fig 3B)? Does this mean that the WT-PGAM5 is not cleaved?

-PGAM5 is cleaved at amino acid residue Ser24. The approximate size of PGAM5 WT GFP full length form is 60 kDa and the cleaved form is around 56 kDa. These bands could not be resolved in 10% or 12% SDS-PAGE gels (which are primarily used in this paper). Similarly, the size of the OMM GFP 58 kDa (full length form) and Δ MM WT mutant is 52 kDa (full length form), so resolution of this size difference was not clear.

Therefore, to test if PGAM5 WT GFP is cleaved, we have run the U2OS cell lysates overexpressing PGAM5 WT GFP in 4-20% gradient gel. PGAM5 WT GFP showed a cleavage pattern similar to endogenous PGAM5 (Supplementary Fig 4A).

6. While the WT PGAM5 can form dimer and dodecamer forms, and the MM PGAM5 can only form dimers, I'm not sure that they can conclude the preference of these forms to bind MFN2/DRP1 based on their data. Are there other explanations? Perhaps the MM deletion impacts binding.

- We observed that Δ MM WT shows higher affinity towards DRP1 in CCCP treated condition even compared to WT GFP (full length). Additionally, Δ MM WT shows higher cytosolic distribution than WT (Fig 1G and 2A). So, we conclude that PGAM5 dimeric form and cytosolic distribution together regulate DRP1 binding.

7. I have some concerns about the quantification of the mitochondrial morphology. The authors use aspect ratio, which is typically a comparison of the length:width ratio of any particular mitochondrial fragment (where the length is defined as the longest axis and the width as the shortest axis). However, the values for the aspect ratio seem to vary between \sim 0.5 to 3. How do you get a ratio below 1 - wouldn't that mean the length is no longer the longest axis and there is something wrong with how the value is calculated. Conversely, if length is the longest axis, how do you get a ratio above 1? Would average mitochondrial length be more appropriate in the context of looking at mitochondrial fragmentation. It is also not clear from the methods and figure legends what kind of replicates were used. The methods state that 60-100 cells were analyzed, but in the figure legends it says n=3. Does n=3 refer to 3 mitochondria, 3 cells or 3 replicates of 60-100 cells. Are the replicates technical or biological.

- Previous literatures have quantified mitochondrial morphology in multiple ways (<https://www.frontiersin.org/journals/physics/articles/10.3389/fphy.2022.855775/full>). The mitochondria aspect ratio in this study is defined as density of mitochondria networks / density of mitochondria puncta. A classifier was manually trained to classify mitochondria in binary images as either networks or puncta (as now explained in the methods section). Networks are defined as large, elongated structures whereas puncta are defined as small circular structures (i.e fragmented mitochondria). We have previously published a detailed protocol for this quantification in this article "Protocol for evaluating mitochondrial morphology changes in response to CCCP-induced stress through open-source image processing software" (reference no. 32). We have also added an explanation of the quantification protocol in detail in the methods section.

The experimental replicates are technical replicates. Three technical replicates have been done for each experiment (n=3). The total number of cells counted combined in all technical replicates are between 60-100.

8. For the mitochondrial fractionation (Sup Fig 3 C/D) there is nothing to indicate how clean the fractions are. It is also not clear why actin was used as a load control for the mitochondrial fraction. Without the proper controls, conclusions from this experiment are meaningless. The authors argue PGAM WT was more cytosolic compared to cleavage resistant forms, which may be true. However, if you look at the cytosolic fraction of PGAM WT in DMSO -treated cells, there does not appear to be any difference in compared to CCCP treatment. Wouldn't the model predict there should be more cytosolic PGAM5 WT with CCCP?

-An extra Western blot has been added representing the mitochondrial fraction (IB: VDAC) of PGAM5 WT in presence and absence of CCCP (Supplementary fig 5D), which confirms our conclusion.

9. The quantification of the starvation induced effects on mitochondrial morphology (Fig 4A/B) appears to be incomplete. Panels C/D only shows the quantification during starvation, and not in normal media. The authors should also show the morphology under normal media, so that we can see if the treatment is inducing mitochondrial hyperfusion as expected - the representative images don't necessarily show this is the case (e.g. mitos in non-transfected cells look the same in normal media and under starvation).

-After quantification of mitochondrial aspect ratio in the control subset under media and starvation condition, we found a moderate increase which is not statistically significant. So, we have adjusted the words in the text accordingly. The represented image of the cells (of the non-transfected control) has been changed in the Figure 4a and b.

10. Authors make a hypothesis about the phosphorylation status of Drp1. This should be very easy to check in their cell lines, as there are commercially available antibodies they are very commonly used. Adding this data would greatly strengthen the conclusions of their model.

- We have checked the DRP1 phosphorylation level in PGAM5 knockdown U2OS cells and the resultant data has been added in Figure 5.

Minor Issues:

11. The model in Fig 3G needs work. In Panel 1, PGAM5 is shown as a dimer, not a dodecamer. In Panel 2, PGAM5 is just as close to MFN2 as in Panel 1 (should be more distant if it dissociates). It is also not clear why the proteasome is depicted in Panel 2, this is not discussed anywhere in the text.

-We have updated the model.

12. The term 'stress' is used throughout the manuscript. However, given that different stresses (CCCP vs Starvation) seem to have distinct effects on PGAM5, the authors should specify which stress they are referring to at all times.

-We have updated the text.

13. I'm not sure I'd agree that starvation-dependent morphology is mostly dependent on DRP1 activity (line 168). See <https://pubmed.ncbi.nlm.nih.gov/19360003/>.

-We agree that starvation-dependent mitochondrial morphology is dependent on multiple factors including L-OPA1, MFN1, SLP-2 and DRP1. We have updated the text accordingly and added this paper as a reference (Reference no. - 28).

14. Sup Fig 4A is labelled a MFN2 binding, but in the main text is referred to as DRP1 binding.

-We have updated the text.

15. Graph axis in Sup Fig 4B is a bit misleading, be sure to clearly indicate the axis break.

- We have explained it in the figure legends.

Reviewer 2:

16. While I believe this study is well-performed, I feel several aspects of the model rely too heavily on inferences from previous papers, or on assumptions. For example, the authors do not directly demonstrate that stress changes the oligomeric state of PGAM5, or directly demonstrate preferential binding of MFN2 or Drp1 to either of these forms. They also make the assumption that PGAM5 controls Drp1 activity through dephosphorylation of Drp1 at S616 - however, at no point do they monitor Drp1 phosphorylation in their assays. A key finding from their paper is that cleaved PGAM5 binds Drp1 in the cytosol. If this is the case, then Drp1 should preferentially interact with

the cleaved (smaller) form of PGAM5. The authors could potentially perform IPs of Drp1 and examine whether it preferentially immunoprecipitates cleaved PGAM5.

-We thank the reviewer for their comments and have added three pieces of data to help clarify.

1. Native-PAGE showing the change in Oligomeric state of PGAM5 after CCCP treatment in U2OS cells (Supplementary fig 3C).
2. DRP1 WT Flag interaction with PGAM5 upon CCCP treatment (Supplementary fig 4).
3. DRP1 Serine phospho616 and 637 levels in PGAM5 knockdown cells upon starvation (Figure 5).

17. A question that also springs to mind is which of these two mechanisms of action (changed oligomeric state versus cleavage) are most important for Drp1 activation. Since Drp1 appears to bind poorly to mitochondrially-localized PGAM5, is the switch in oligomeric state only a minor contributor to PGAM5's regulation of Drp1 activity, with the main role, at least under stress conditions, being through release of the cleaved fragment?

-We agree with the reviewer. We also believe that cytosolic release of PGAM5 plays the most essential role in binding DRP1, and change in oligomeric state (dimer formation) probably potentiates the process.

18. Minor point - It would be helpful to have more detail about how the mitochondrial morphology analysis was performed. How is the 'Aspect ratio' determined?

-Reviewer 1 also had this comment and therefore we have added the method of aspect ratio quantification in the materials and methods section.

Second decision letter

MS ID#: jcs.263903R1

MS TITLE: PGAM5 cleavage and oligomerization equilibrates mitochondrial dynamics under stress by regulating DRP1 function

AUTHORS: Sudeshna Nag; Kaitlin Szederkenyi; Christopher M. Yip; Angus McQuibban

ARTICLE TYPE: Research Article

Dear Dr McQuibban,

We have now reached a decision on the above manuscript.

To see the reviewers' reports and a copy of this decision letter, please go to: *****

As you will see, the reviewers gave favourable reports but one of them has raised some final points that, I feel, will require amendments to your manuscript. I hope that you will be able to deal with these points (ideally experimentally but otherwise with text changes to acknowledge the limitations of the relevant experiments) because I would like to be able to accept your paper on resubmission.

Second revision

Author response to reviewers' comments

- With regards to the fractionation studies, please could you change the wording where applicable to mention that the proteins were present in the "crude mitochondrial fraction" and/or the "supernatant", i.e., instead of claiming that the proteins are mitochondrial or not.
 - We have changed the text accordingly (line number 175-178).
 - As highlighted by one of the reviewers (Reviewer 1, major point 1), the protein appears in *both* fractions, so this should also be acknowledged in the text.
 - We have changed the text accordingly (line number 175-178).
-

Third decision letter

MS ID#: jcs.263903R2

MS TITLE: PGAM5 cleavage and oligomerization equilibrates mitochondrial dynamics under stress by regulating DRP1 function

AUTHORS: Sudeshna Nag; Kaitlin Szederkenyi; Christopher M. Yip; Angus McQuibban

ARTICLE TYPE: Research Article

Dear Dr McQuibban,

Many thanks for submitting a revised version of your article to JCS. I have now had the chance to assess the revised article and data, and your response to reviewers, and I am pleased to let you know that I feel that the article is now suitable for publication. However, given the comments from the reviewers in the previous round of review I still feel that some very minor additional changes to the text are required:

- With regards to the fractionation studies, please could you change the wording where applicable to mention that the proteins were present in the "crude mitochondrial fraction" and/or the "supernatant", i.e., instead of claiming that the proteins are mitochondrial or not.
- As highlighted by one of the reviewers (Reviewer 1, major point 1), the protein appears in both fractions, so this should also be acknowledged in the text.

Once this is done, the article can be accepted.

Fourth decision letter

MS ID#: jcs.263903R3

MS Title: PGAM5 cleavage and oligomerization equilibrates mitochondrial dynamics under stress by regulating DRP1 function

Authors: Sudeshna Nag; Kaitlin Szederkenyi; Christopher M. Yip; Angus McQuibban

Article Type: Research Article

Dear Dr McQuibban,

Thank you for making those final changes to your article and resubmitting an updated version. I am happy to tell you that your manuscript has now been accepted for publication in Journal of Cell Science, pending standard publication integrity checks.